



# Influence of the Change in Total Ozone Column (TOC) on the Occurrence of Tropospheric Ozone Depletion Events (ODEs) in the Antarctic

Le Cao[1,*], Linjie Fan[1,2,*], Simeng Li[1], and Shuangyan Yang[2]

[1]Key Laboratory for Aerosol-Cloud-Precipitation of China Meteorological Administration, Nanjing University of Information Science and Technology, Nanjing 210044, China
[2]Key Laboratory of Meteorological Disaster, Ministry of Education (KLME)/Joint International Research Laboratory of Climate and Environmental Change (ILCEC)/Collaborative Innovation Center on Forecast and Evaluation of Meteorological Disasters (CIC-FEMD), Nanjing University of Information Science and Technology, Nanjing 210044, China
[*]These authors contributed equally to this work.

**Correspondence:** L. Cao
(le.cao@nuist.edu.cn)

**Abstract.** The occurrence of the tropospheric ozone depletion events (ODEs) in the Antarctic can be influenced by the change in Total Ozone Column (TOC). In this study, we combined the observational data obtained from ground observation stations with two numerical models (TUV and KINAL), to figure out the relationship between the TOC change and the occurrence of ODEs in the Antarctic. A sensitivity analysis was also performed on the change in ozone and major bromine species (BrO, HOBr and HBr) to find out key photolysis reactions determining the impact on the occurrence of tropospheric ODEs brought by the change in TOC. From the analysis of the observational data and the numerical results, we suggested that the occurrence frequency of ODEs in the Antarctic seems negatively correlated with the variation of TOC. Moreover, major ODE accelerating reactions (i.e. photolysis of ozone, $H_2O_2$ and HCHO) and decelerating reactions (i.e. photolysis of BrO and HOBr), which heavily control the start of ODEs, were also identified. It was found that when TOC varies, the major ODE accelerating reactions speed up significantly, while major ODE decelerating reactions are only slightly affected, thus leading to the negative dependence of the ODE occurrence on the change in TOC.

## 1 Introduction

Ozone is a short-lived trace gas in the atmosphere, with about 90% located in the stratosphere and 10% in the troposphere (Seinfeld and Pandis, 2006; Akimoto, 2016). In the stratosphere, ozone plays a role in absorbing the ultraviolet (UV) radiation from the sun, thus protecting the lives on the earth. In contrast, ozone in the troposphere is a pollutant. It causes eye irritations and disorders of the lung function of human beings at a high concentration (Lippmann, 1991) Moreover, ozone in the troposphere also acts as a greenhouse gas, contributing to the global warming (Seinfeld and Pandis, 2006). It was suggested by Toumi et al. (1996) that the tropospheric ozone mostly originates from three sources: downward entrainment from the strato-





sphere, photochemical reactions occurring in the troposphere, and the vertical convection. Thus, the amount of ozone in the
troposphere can be affected by many factors such as the variation of the stratospheric ozone.

Ozone in polar regions is always a focus of the scientific community. Due to the special geographical location and the unique environment, polar regions are also called "natural laboratory" of the earth (Heinemann, 2008). Moreover, because polar regions especially the Antarctic are hardly affected by anthropogenic activities, the climate of polar regions is capable of reflecting the global change of the climate (Prather and Jaffe, 1990). In 1980s, a extraordinary event, i.e. ozone hole,
was found occurring over the Antarctic (Farman et al., 1985) . This event refers to a continuous decline in the total ozone amount over the Antarctic during the springtime of every year. Because the majority of ozone in the atmosphere resides in the stratosphere, the Antarctic ozone hole mostly represents a depletion of the stratospheric ozone. After the discovery of the ozone hole, large efforts were made to reveal the reasons causing the emergence of this event, such as figuring out the role of chlorofluorocarbons (CFCs) from human activities (Molina and Rowland, 1974; Bedjanian and Poulet, 2003), heterogeneous
reactions on the surface of PSCs (polar stratospheric clouds) and the photolysis of ClO dimer, i.e., ClOOCl at polar night (Finlayson-Pitts and Pitts, 1999; Brasseur and Solomon, 2005).

Similar to the ozone hole phenomenon representing a depletion of the stratospheric ozone, in the 1980s, an ozone depletion event (ODE) was also observed in the troposphere of polar regions (Oltmans, 1981). It was firstly reported that during the springtime of Arctic, the surface ozone drops from a normal level ($\sim$40 ppb) to less than 1 ppb within a few hours or 1-2 days.
After that, the tropospheric ODE was also reported occurring in coastal areas of Antarctic (Kreher et al., 1997; Frieß et al., 2004; Wagner et al., 2007). Subsequent studies suggested that the tropospheric ODE is a common phenomenon that occurs in the atmospheric boundary layer during the springtime of both the Arctic and the Antarctic. It was also reported by Roscoe and Roscoe (2006) that the tropospheric ODE occurs in the Antarctic since as early as the 1950s. Following studies suggested that the occurrence of the tropospheric ODE is related to an auto-catalytic reaction cycle involving bromine species, as follows
(Simpson et al., 2007):

$$\mathrm{Br_2} + h\nu \rightarrow 2\mathrm{Br},$$
$$\mathrm{Br} + \mathrm{O_3} \rightarrow \mathrm{BrO} + \mathrm{O_2},$$
$$\mathrm{BrO} + \mathrm{HO_2} \rightarrow \mathrm{HOBr} + \mathrm{O_2},$$
$$\underline{\mathrm{HOBr} + \mathrm{H^+} + \mathrm{Br^-} \overset{\mathrm{mp}}{\rightarrow} \mathrm{Br_2} + \mathrm{H_2O},}$$
$$\mathrm{Net:} \mathrm{O_3} + \mathrm{HO_2} + \mathrm{H^+} + \mathrm{Br^-} + h\nu \overset{\mathrm{mp}}{\rightarrow} 2\mathrm{O_2} + \mathrm{Br} + \mathrm{H_2O}. \tag{I}$$

This bromine-involved reaction cycle includes heterogeneous reactions occurring on substrates such as the snow-/ice-covered ground surface and the suspended aerosols. Through reaction cycle (I), bromide ions ($\mathrm{Br^-}$) are activated from the substrates, and the total bromine amount in the troposphere is thus elevated. The bromine released to the atmosphere then consumes
ozone near the ground, leading to the occurrence of the tropospheric ODEs. This ozone-depleting cycle is thus called "bromine explosion mechanism" (Platt and Janssen, 1995; Platt and Lehrer, 1997; Wennberg, 1999).

Apart from the bromine chemistry, the occurrence of the tropospheric ODE was also found determined by many factors such as: (1) Temperature. Tarasick and Bottenheim (2002) examined historical ozonesonde records at three Canadian stations



over the time period 1966-2000. They suggested that a low temperature (<-20°C) is probably a necessary condition for the occurrence of ODEs, because heterogeneous reactions that activate bromide from substrates and the formation of frost flowers are favored under this cold condition. However, in a later analysis of ozone data obtained from a transpolar drift, Bottenheim et al. (2009) found the temperature well above -20°C during the most persistent ozone depletion period over the Arctic Ocean. It was also reported by Koo et al. (2012) that there is no evidence in observations for the threshold value of temperature for the occurrence of ODEs. Instead, they suggested the variability of temperature potentially an important factor for the depletion of ozone. (2) Passing of pressure systems. By analyzing the values of ozone mixing ratio and meteorological parameters from balloon sondes during the 1994 Polar Sunrise Experiment (PSE94), Hopper et al. (1998) suggested that the occurrence of ODEs in the Arctic is strongly correlated with high-pressure systems. This dependence of the ozone decline on pressure systems was also confirmed by Jacobi et al. (2010) who proposed that mesoscale synoptic systems are able to transport air masses with low ozone mixing ratio to the observational site, leading to the detection of ODEs at Arctic coastal stations. It was also suggested by Boylan et al. (2014) that the transport caused by synoptic patterns acts as the major factor for the occurrence of ODEs at Barrow, Alaska, rather than the change in local meteorological parameters. In contrast, Jones et al. (2006) analyzed the observational data of ozone and meteorological parameters obtained at Halley station in coastal Antarctica, and they figured out that in Antarctica, the occurrence of ODEs is highly associated with low pressure systems, denoting the remarkable differences in the atmospheric system between the Arctic and the Antarctic. (3) Formation of fresh sea ice. Based on sea ice maps obtained from satellite detection, Bottenheim et al. (2009) figured out regions of the Arctic Ocean as the origin of the tropospheric ozone depletion. In these regions, open leads, polynyas and fresh sea ice are frequently formed, which favors the release of bromine and thus the depletion of ozone. These ODE-originating regions proposed by Bottenheim et al. (2009) are also consistent with the "cold spots" discovered in a previous study of Bottenheim and Chan (2006) where the depletion of ozone possibly initiates and develops. The connection between the ODE occurrence and the formation of fresh sea ice was also identified by Jones et al. (2006) by revealing that the air masses causing rapid ozone depletion at Halley station originate in a region where a large amount of fresh sea ice is formed. (4) Other factors. ODEs were also found impacted by the presence of mixed-phase clouds in the boundary layer due to the cloud-top radiative cooling (Hu et al., 2011) and regional climate variability such as the Western Pacific (WP) teleconnection pattern (Koo et al., 2014).

Although there exist many studies discussing the determining factors for the occurrence of the tropospheric ODE, the impact caused by the change in the stratospheric ozone on the occurrence of the tropospheric ODE has not been thoroughly investigated yet. In previous studies, most of often, ozone change in the stratosphere and the tropospheric ODEs were investigated separately. However, ODEs can be strongly influenced by the change in the stratospheric ozone. For example, the ozone-lacking air in the boundary layer can be partly replenished by the ozone-rich air due to the downward entrainment from the lower stratosphere (Kuang et al., 2017). However, because of the strong stability of the polar boundary layer during ODEs, this influence might be minor. Moreover, the variation of the stratospheric ozone would also lead to a change in the solar radiation reaching the ground surface, thus affecting the rates of photolysis reactions in the troposphere. As a result, the lifetimes of many atmospheric constituents in the troposphere and the occurrence frequency of the tropospheric ODEs can be altered. But it is still unclear whether the change in the stratospheric ozone will foster or retard the occurrence of the tropospheric ODE.



Therefore, in the present study, we combined the observational data from ground observation stations with two numerical models, to figure out the impacts on the occurrence of tropospheric ODEs in the Antarctic brought about by the change in the total ozone amount including the stratospheric ozone. A concentration sensitivity analysis was also performed to reveal photolysis reactions heavily responsible for this impact on the tropospheric ODE.

The structure of the manuscript is as follows. In Sect. 2, observational data analyzed in the present study are described. The numerical models and the governing equations are also presented in this section. In Sect. 3, results obtained from the analysis of the observational data and the computations of the numerical models are shown and discussed. At last, in Sect. 4, conclusions achieved in this study are summarized, and studies to be made in the future are also prospected.

## 2    Description of the Observational Data and the Numerical Method

In the present study, we first analyzed the observational data of ozone obtained from ground observation stations, and then tried to figure out the relationship between the variation of the total ozone column (TOC) and the occurrence frequency of tropospheric ODEs. Then we took the Halley station as an example, and applied two numerical models, TUV model and KINAL box model, to capture the temporal change in ozone and major bromine species during ODEs under the conditions of the Halley station. Sensitivity tests were also performed in these models to figure out the influence caused by the change in the total ozone column (TOC) on the time variation of these air constituents (i.e. ozone and bromine species). Then the most influential photolysis reactions were identified through a concentration sensitivity analysis on the change in these air constituents.

### 2.1    Observational Data of Ozone

Two different types of ozone data were used in the present study. One is the total ozone column (TOC) obtained from ground-based observations. This kind of data mainly indicates the total ozone amount in a vertical column extending from the ground surface to the top of the atmosphere. Because approximately 90% of ozone in the atmosphere resides in the stratosphere, the change in TOC can heavily reflect the variation of the stratospheric ozone. The other type of the ozone data is the near-surface ozone mixing ratio recorded at ground observation stations, and it can partly represent the ozone concentration in the atmospheric boundary layer. The details of these ozone data are given below.

### 2.1.1    Total Ozone Column (TOC)

The TOC data used in this study were obtained from World Ozone and Ultraviolet Radiation Data Center of Canada (WOUDC, https://woudc.org/home.php) for all the registered stations in the Antarctic. These TOC data were observed using a Dobson instrument (Balis et al., 2007), and cover a time span from the year 2000 to 2016. The time resolution of these data is 1 day. After filtering out stations that possess only out-dated or incomplete data, we picked up Halley station (75.52°S, 26.73°W), which has the most complete TOC data, as an example to indicate the typical temporal change in TOC in the Antarctic. In order to guarantee the representativeness of the data from this station, we compared the time series of TOC recorded at the Halley





station with those obtained at many other stations in Antarctic (see Fig. 1). It can be seen that the trend of TOC at the Halley station is analogous to those obtained at other observation stations. Furthermore, the correlation coefficients between the TOCs obtained at the Halley station and other stations mostly possess a value above 0.7 (shown in Tab. S1 of the supplements), indicating that the TOC data obtained at the Halley station is significantly correlated with those obtained at other stations, and thus can represent a typical TOC change in the Antarctic. The low correlation coefficient between the Halley station and the Faraday-Vernadsky station might be caused by the special geographical location of the Faraday-Vernadsky station, which is out of the scope of the present study.

### 2.1.2 Surface Ozone

After choosing the Halley station as an example representing the typical TOC change in the Antarctic, we adopted the surface ozone data at the Halley station from World Data Center for Greenhouse Gases (WDCGG, https://gaw.kishou.go.jp), and the time span of the surface ozone data is between the year 2007 and 2013. The time resolution of the surface data is 1 hour. Because the tropospheric ODE, which we focused on in the present study, mostly occurs in the springtime of every year, we thus only analyzed the surface ozone data during the springtime of the Halley station (from Sept. 1 to Nov. 30).

We then calculated the occurrence frequency of the tropospheric ODE at the Halley station for each month based on the daily data of the surface ozone. The occurrence frequency of the ODE, $f$, is estimated as follows,

$$f = N_{\mathrm{ODEs}}/N_{\mathrm{total}}, \tag{1}$$

in which $N_{\mathrm{ODEs}}$ is the number of time points that ODE occurs, and $N_{\mathrm{total}}$ is the total number of time points in this month. However, the definition of the ODE occurrence is still under debate. In some previous studies (Tarasick and Bottenheim, 2002; Koo et al., 2012), the occurrence of ODEs was recognized by the surface ozone mixing ratio. When the surface ozone drops to lower than 20 ppb, it is called partial ODEs. Moreover, when the surface ozone declines to a level lower than 10 ppb, it is called major or severe ODEs. In contrast, the ODEs can also be judged by the change in the ozone mixing ratio. This method was suggested by Bian et al. (2018) to indicate extreme events in polar regions. In the present study, we used the method suggested by Bian et al. (2018) and picked up the time point representing the occurrence of ODEs from the observational data when it fulfills the following criterion,

$$[O_3]_i - \overline{[O_3]} < -2\sigma, \tag{2}$$

where $[O_3]_i$ is the instantaneous ozone data at the $i$-th time point, and $\overline{[O_3]}$ is the mean ozone value of this month. $\sigma$ in Eq. (2) is the standard deviation. After picking up the time points representing the tropospheric ODEs from the observational data, we calculated the frequency of ODEs belonging to each month during the springtime of the years 2007-2013.

### 2.2 Numerical Methods

Two numerical models, a radiation model (TUV, Tropospheric Ultraviolet and Visible) (Madronich and Flocke, 1997, 1999) and a chemical box model (KINAL, KInetic aNALysis of reaction mechanics) (Turányi, 1990), were used in this study. The



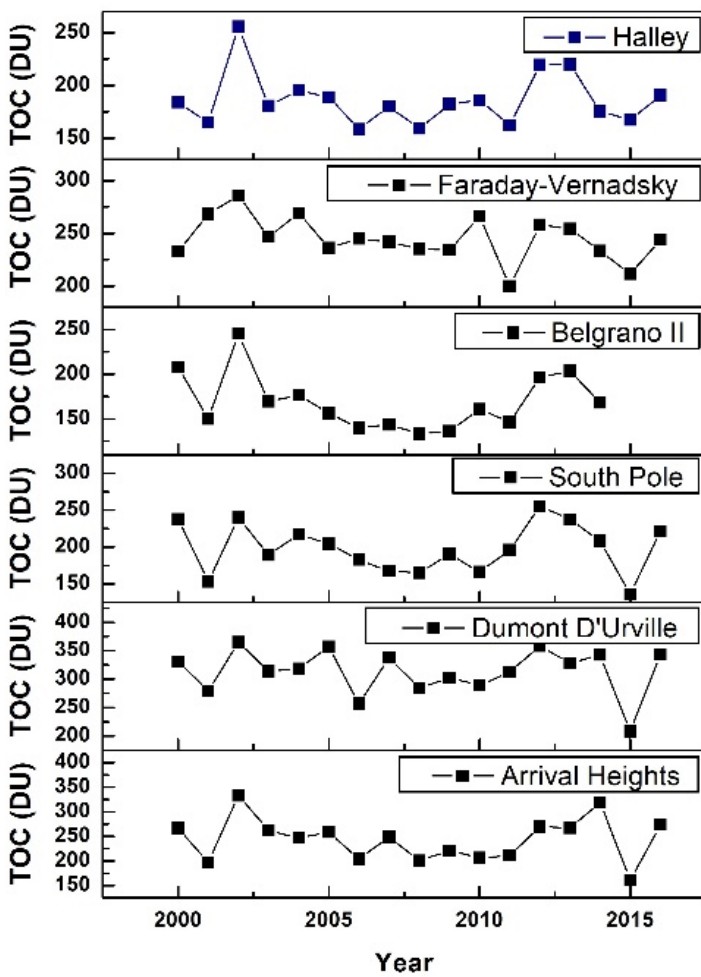

**Figure 1.** The temporal change in TOCs observed at Halley, Faraday-Vernadsky, Belgrano II, South Pole, Dumont D'Urville, and Arrival Heights stations from the year 2000 to 2016.

TUV model was used to estimate the photolytic rates of atmospheric constituents in the troposphere of the Antarctic. The KINAL box model was used to capture the concentration change in these constituents such as ozone in the boundary layer over time. KINAL was also used to compute the sensitivity of these constituents to each photolysis reaction in the chemical mechanism.





### 2.2.1 TUV model

The TUV model (Tropospheric Ultraviolet and Visible Radiation Model), provided by NCAR (National Center for Atmospheric Research), is able to calculate the tropospheric photo-dissociation coefficient (Madronich and Flocke, 1997, 1999), based on input parameters such as the total ozone column (TOC). The vertical ozone profile assumed in the model is taken from the US standard atmosphere. A total of 112 photolysis reactions are implemented in the TUV model.

The photolytic rate constant $j_p$ (unit: s$^{-1}$) for each photolysis reaction is calculated by the TUV model as follows,

$$j_p = \int\limits_0^\infty \sigma(\lambda)\Phi(\lambda)F(\lambda)\mathrm{d}\lambda. \tag{3}$$

In Eq. (3), $\sigma(\lambda)$ represents the absorption cross section at the wavelength $\lambda$. $\Phi(\lambda)$ denotes the photolytic quantum yield. $F$ in Eq. (3) is the actinic flux, and it is determined by many factors such as the presence of clouds and the change in TOC. A 4-stream discrete ordinate method (van Oss and Spurr, 2002) with a step length of 1 nm is implemented in TUV, calculating the photolytic rate constants.

Thus, in the present study, we implemented the observed TOC and other weather conditions into the TUV model, to estimate the actinic flux $F$ reaching the boundary layer and the rates of photolysis reactions for different time periods. Then we used a chemical box model, KINAL, to capture the concentration change in ozone and bromine species in the process of the tropospheric ODE under different photolytic conditions. By doing that, the influence of the change in the total amount of ozone, i.e. TOC, on the occurrence of the tropospheric ODE can be revealed.

### 2.2.2 KINAL model

After obtaining the photolytic dissociation rates of many atmospheric constituents using the TUV model, we then applied the chemical box model, KINAL (KInetic aNALysis of reaction mechanics) (Turányi, 1990) to capture the temporal evolution of chemical species such as ozone and many bromine species. Moreover, sensitivities of these chemical species to each photolysis reaction in the chemical mechanism were also computed using KINAL.

The governing equation describing the temporal evolution of chemical species in the KINAL model is as follows (Turányi, 1990):

$$\frac{\mathrm{d}\boldsymbol{c}}{\mathrm{d}t} = \boldsymbol{f}(\boldsymbol{c},\boldsymbol{k}) + \boldsymbol{E}, \tag{4}$$

with the initial condition $\boldsymbol{c}|_{t=0} = \boldsymbol{c}_0$, where $\boldsymbol{c}$ represents a vector of chemical species concentrations. $\boldsymbol{k}$ in Eq. (4) is a vector of rate constants of chemical reactions and $t$ denotes time. $\boldsymbol{E}$ indicates the near-surface source emissions, and $\frac{\mathrm{d}\boldsymbol{c}}{\mathrm{d}t}$ is the temporal evolution of chemical species such as ozone. In the present study, a chemical mechanism including the bromine and chlorine chemistry was adopted from previous box model studies (Cao et al., 2014, 2016a,b; Zhou et al., 2020), and the reaction rate constants were updated with the latest chemical kinetic data (Atkinson et al., 2006). There are in total 49 species and 141 chemical reactions included in the latest version of the chemical mechanism, which are listed in Tab. S2 of the supplementary material. Among these reactions, there are 23 photolysis reactions of which the rates are associated with the change in TOC,





**Table 1.** Listing of photolysis reactions in the chemical mechanism of the KINAL model, of which the rates vary with TOC. The reaction numbers correspond to those listed in Tab. S2 of the supplements.

| Reaction Number | Reaction |
|---|---|
| (SR1) | $O_3 + h\nu \rightarrow O(^1D) + O_2$ |
| (SR6) | $Br_2 + h\nu \rightarrow 2Br$ |
| (SR7) | $BrO + h\nu \xrightarrow{O_2} Br + O_3$ |
| (SR11) | $HOBr + h\nu \rightarrow Br + OH$ |
| (SR57) | $H_2O_2 + h\nu \rightarrow 2OH$ |
| (SR58) | $HCHO + h\nu \xrightarrow{2O_2} 2HO_2 + CO$ |
| (SR59) | $HCHO + h\nu \rightarrow H_2 + CO$ |
| (SR60) | $C_2H_4O + h\nu \rightarrow CH_3O_2 + CO + HO_2$ |
| (SR61) | $CH_3O_2H + h\nu \rightarrow OH + HCHO + HO_2$ |
| (SR62) | $C_2H_5O_2H + h\nu \rightarrow C_2H_5O + OH$ |
| (SR74) | $HNO_3 + h\nu \rightarrow NO_2 + OH$ |
| (SR75) | $NO_2 + h\nu \xrightarrow{O_2} NO + O_3$ |
| (SR76) | $NO_3 + h\nu \xrightarrow{O_2} NO_2 + O_3$ |
| (SR77) | $NO_3 + h\nu \rightarrow NO + O_2$ |
| (SR88) | $BrONO_2 + h\nu \rightarrow NO_2 + BrO$ |
| (SR89) | $BrNO_2 + h\nu \rightarrow NO_2 + Br$ |
| (SR91) | $PAN + h\nu \rightarrow NO_2 + CH_3CO_3$ |
| (SR135) | $BrCl + h\nu \rightarrow Br + Cl$ |
| (SR136) | $Cl_2 + h\nu \rightarrow 2Cl$ |
| (SR137) | $ClO + h\nu \rightarrow Cl + O_3$ |
| (SR138) | $HOCl + h\nu \rightarrow Cl + OH$ |
| (SR139) | $ClONO_2 + h\nu \rightarrow Cl + NO_3$ |
| (SR140) | $OClO + h\nu \rightarrow ClO + O_3$ |

and these photolysis reactions are listed in Tab. 1 along with the reaction numbers in the mechanism. Among these photolysis reactions, a part of them can enhance the occurrence of ODEs, while the others will retard it.

In the KINAL model, it is also assumed that bromide stored in the ice/snow-covered ground surface is inexhaustible. As a result, the rates of heterogeneous reactions such as $HOBr_{(gas)} + H^+_{(liquid)} + Br^-_{(liquid)} \rightarrow Br_{2(gas)} + H_2O_{(liquid)}$, which are responsible for the bromine explosion, only depend on the availability of HOBr in the atmosphere in model calculations. Moreover, the wind speed was assumed as $8\,m\,s^{-1}$ (Beare et al., 2006), which is a typical wind speed in polar regions. In addition, the roughness of the ice/snow surface is set to $10^{-5}\,m$ (Stull, 1988), and the height of the polar boundary layer is





**Table 2.** Initial atmospheric composition in the boundary layer of the Antarctic (ppm = parts per million, ppb = parts per billion, ppt = parts per trillion) (Piot, 2007), and the prescribed intensities of emission fluxes from the ground surface (units: molec. cm$^{-2}$ s$^{-1}$) (Hutterli et al., 2004; Riedel et al., 2005; Jones et al., 2011), assumed in the model.

| Species | Mixing Ratio | Emissions | Species | Mixing Ratio | Emissions |
|---------|--------------|-----------|---------|--------------|-----------|
| $O_3$ | 25 ppb | - | $C_2H_6$ | 0.4 ppb | - |
| $Br_2$ | 0.3 ppt | - | $C_2H_4$ | 50 ppt | - |
| HBr | 0.01 ppt | - | $C_2H_2$ | 300 ppt | - |
| $CH_4$ | 1.7 ppm | - | $C_3H_8$ | 0.2 ppb | - |
| $CO_2$ | 371 ppm | - | NO | 2 ppt | $1.6 \times 10^7$ |
| CO | 50 ppb | - | $NO_2$ | 8 ppt | $1.6 \times 10^7$ |
| HCHO | 500 ppt | $9.0 \times 10^9$ | HONO | - | $1.6 \times 10^7$ |
| $CH_3CHO$ | 500 ppt | - | $H_2O_2$ | - | $1.0 \times 10^9$ |
| $H_2O$ | 800 ppm | - | | | |

assumed as 200 m because the typical thickness of the boundary layer in polar regions is about 100-500 m (Simpson et al., 2007; Anderson and Neff, 2008). The initial atmospheric composition used in the model is listed in Tab. 2, which represents a

190    typical air composition in the Antarctic (Piot, 2007). Constant emission fluxes from the ground surface were also prescribed in the model according to observations (Hutterli et al., 2004; Riedel et al., 2005; Jones et al., 2011), and the prescribed intensities of the emission fluxes are also presented in Tab. 2.

### 2.2.3    Concentration Sensitivity Analysis

After obtaining the temporal evolution of ozone and major bromine species, relative concentration sensitivities of these species

195    to different photolysis reactions in the chemical mechanism were computed to reveal the dependence of these species on each photolysis reaction of the mechanism. The relative concentration sensitivity $S_{ij}$ is calculated by:

$$S_{ij} = \frac{\partial \ln c_i}{\partial \ln k_j} = \frac{k_j}{c_i} \frac{\partial c_i}{\partial k_j}, \tag{5}$$

which shows the importance of the $j$-th reaction for the concentration change in the $i$-th chemical species. In Eq. (5), $i$ is the index of chemical species, and $j$ is the index of chemical reactions in the mechanism. $c_i$ is the concentration of the $i$-th species,

200    and $k_j$ is the rate constant of the $j$-th reaction. $S_{ij}$, an element of the relative sensitivity matrix, indicates the change in the $i$-th species concentration resulted from a small perturbation in the $j$-th reaction rate. The evaluation of the concentration sensitivity is helpful for discovering the importance of specific reactions in the chemical mechanism for the concentration change in the focused species.

In the following section, the computational results are presented and discussed.





## 3   Results and Discussions

In this section, we first show the relationship between the TOC change and the occurrence frequency of the tropospheric ODE at the Halley station based on the observational data. Later, we presented the computational results of ODEs for the year 2013 as an example to show the time variations of ozone and major bromine compounds during ODEs. The depletion rate of ozone and the temporal change in bromine species under different weather conditions were then displayed to indicate the influence caused by the TOC change on the ozone depletion and the bromine activation. At last, a concentration sensitivity analysis was performed to see which photolysis reactions playing important roles in the connection between the TOC change and the occurrence of ODEs.

### 3.1   Relationship between the TOC Change and the Tropospheric ODE

The time series of the monthly-averaged TOC at the Halley station and the occurrence frequency of ODEs are presented in Fig. 2. In general, it seems that the occurrence of ODEs is negatively correlated with the temporal change in TOC, with a correlation coefficient of -0.27. It is seen that during the springtime of each year at the Halley station, mostly the ODEs occur more frequently in September and October than in November. Meanwhile, TOC was also found increasing from September to November in most of the years investigated (see Fig. 2). However, this negative correlation does not guarantee that the occurrence of ODEs is negatively dependent on the change in TOC, because that apart from the change in TOC, the variation of solar zenith angle (SZA) between different months would also heavily influence the solar radiation reaching the boundary layer, consequently leading to an altering of the ODE occurrence. Thus, in order to clarify it, we ranked the values of TOC and the ODE occurrence frequencies belonging to different months for the years 2007-2013 (see Tab. 3). From the comparison between the ranks of TOC and the ODE occurrence frequencies, it is still difficult to determine the association between the occurrence of ODEs and the variation of TOC. In October, it seems that higher TOC leads to a lower occurrence frequency of ODEs (see Tab. 3). However, this tendency is different in November, in which higher TOC seems to cause a higher occurrence frequency of ODEs. But it should be noted that the occurrence frequency of ODEs in November is remarkably lower than those in September and October. Thus, from the point view of the observational data, despite the small sample size of the ranks of the monthly averaged ozone data and the ODE occurrence frequency, we suggested that the occurrence of ODEs seems negatively dependent on the change in TOC, although the negative correlation is not very evident. To validate this suggestion, more observational data with extended time periods are needed, which unfortunately we are currently lack of.

In order to further clarify the correlation between the change in TOC and the occurrence of ODEs, we then took the year 2013 as an example and used the models (i.e. TUV and KINAL) with the input data of TOC belonging to the year 2013 (220.03 DU on average). The reason we chose the year 2013 is that it is the latest year with abundant springtime surface ozone data.



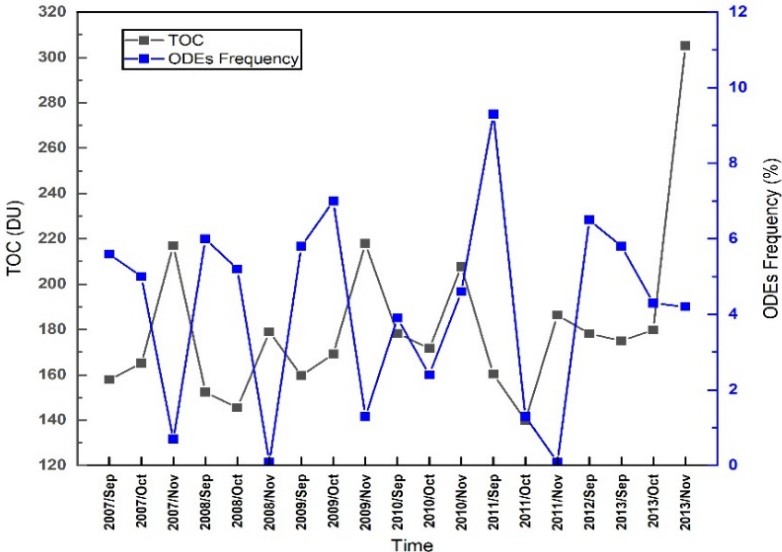

**Figure 2.** Time series of TOC over the Halley station and the occurrence frequency of ODEs during the springtime from the year 2007 to 2013 (the observational data for October and November in the year 2012 are missing).

**Table 3.** Ranks of the monthly averaged TOC (units: DU) and the occurrence frequency of ODEs at the Halley station for the years 2007-2013. Note that the observational data for October and November in the year 2012 are missing.

| Year | Sept. | | | | Oct. | | | | Nov. | | | |
|------|-------|------|-------|------|------|------|-------|------|------|------|-------|------|
|      | TOC   | Rank | Freq. | Rank | TOC  | Rank | Freq. | Rank | TOC  | Rank | Freq. | Rank |
| 2007 | 157.9 | 6    | 0.056 | 6    | 165.1 | 5   | 0.050 | 3    | 217.0 | 4   | 0.007 | 4    |
| 2008 | 152.4 | 7    | 0.060 | 3    | 145.5 | 6   | 0.052 | 2    | 179.0 | 7   | 0     | 6    |
| 2009 | 159.7 | 5    | 0.058 | 5    | 169.2 | 4   | 0.070 | 1    | 218.0 | 3   | 0.013 | 3    |
| 2010 | 178.3 | 1    | 0.039 | 7    | 171.7 | 3   | 0.024 | 5    | 207.7 | 5   | 0.046 | 1    |
| 2011 | 160.3 | 4    | 0.093 | 1    | 139.9 | 7   | 0.013 | 6    | 186.4 | 6   | 0     | 5    |
| 2012 | 178.2 | 2    | 0.070 | 2    | 180.3 | 1   |       |      | 300.2 | 2   |       |      |
| 2013 | 175.0 | 3    | 0.058 | 4    | 179.9 | 2   | 0.043 | 4    | 305.2 | 1   | 0.042 | 2    |

## 235  3.2  Temporal Behavior of Ozone and Bromine Species during ODEs for the Year 2013

The temporal change in ozone and bromine species under the conditions of the year 2013 with an average TOC of 220.03 DU is shown in Fig. 3. From the temporal behavior of these chemical species, we can better understand the interconversion of



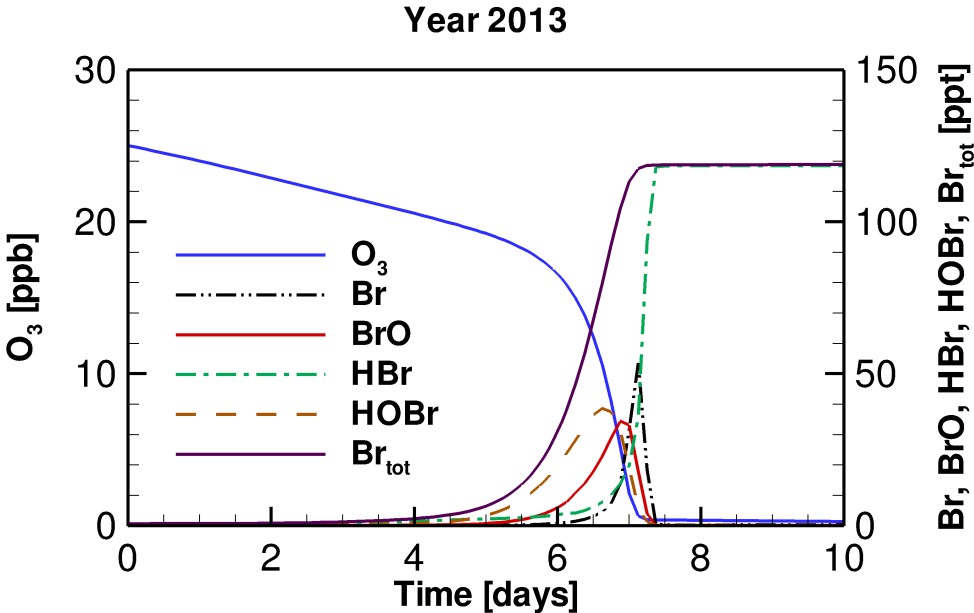

**Figure 3.** Variations of ozone and major bromine species with time during the tropospheric ODE under the conditions of the year 2013.

bromine species and the reasons causing the depletion of ozone in the troposphere. Because similar results have been shown and discussed in previous publications (Cao et al., 2014, 2016a), we only describe it briefly here.

It can be seen in Fig. 3 that the whole process of the ODE can be divided into four time stages, according to the types of bromine species in the troposphere. The first stage is the beginning of ODEs, in which the concentration of bromine is low, and ozone is only slightly depleted. This time stage would last for about 6 days in the present simulation (see Fig. 3). Subsequently, in the second time stage, $Br_2$ released from substrates such as the ice/snow-covered surface due to the bromine explosion mechanism is continuously photolyzed, forming Br atoms. The Br atoms then react with ozone and form BrO:

$$Br + O_3 \rightarrow BrO + O_2. \tag{R1}$$

A part of BrO then gets oxidized and converted to HOBr:

$$BrO + HO_2 \rightarrow HOBr + O_2. \tag{R2}$$

Thus, in the second time stage, the major bromine species are BrO and HOBr. In contrast, the concentration of Br is almost zero due to the presence of ozone. Under this high-HOBr environment, a large amount of bromide is activated from substrates,

so that ozone is rapidly consumed by the massive amount of bromine in the atmosphere. Thus, this second time stage is also the key time period that the majority of the tropospheric ozone is depleted, and this time stage lasts for about 1 day, in which the majority of ozone is depleted with a rate of 0.5-1.0 ppb hr$^{-1}$.





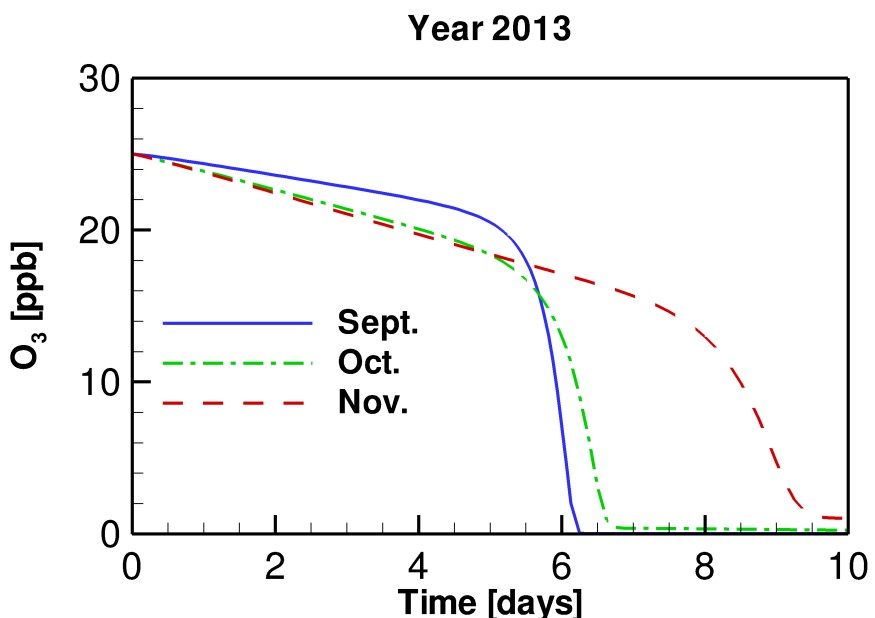

**Figure 4.** Variations of ozone with time during ODEs for different months of the springtime of 2013.

When the mixing ratio of ozone drops to less than 5 ppb, this event enters the third stage. In this time stage, the depletion of ozone continues, but the formations of BrO and HOBr are retarded, because of the low ozone in the atmosphere. The major bromine species at this time is Br, formed by the photo-decomposition of BrO and HOBr in the atmosphere. Then the last stage comes, in which ozone in the troposphere is almost completely consumed. At this time stage, BrO and HOBr are all photolyzed to Br, thus not existing in the atmosphere. Meanwhile, the formed Br is eliminated by aldehydes and $HO_x$ free radicals in the atmosphere, and is converted into HBr. As a result, at this last stage, a complete depletion of ozone in the troposphere is achieved, and the major bromine species is HBr, which is in accordance with previous observations (Langendörfer et al., 1999).

### 3.3 Impact of the Change in TOC on the Occurrence of ODEs

We then compared the model results for different months of the springtime of 2013. The TOC values of these three months, September, October and November, are 175.0, 179.9 and 305.2, respectively. However, it should be noted that aside from the change in TOC, the value of SZA also varies between different months, which may significantly affect the radiation fluxes reaching the boundary layer and thus the rates of photolysis reactions.

The temporal change in ozone during ODEs under the conditions of different months belonging to the year 2013 is shown in Fig. 4. It is seen that compared with the situation in September, the decline of ozone in November is delayed. Moreover, the depletion rate in November was also found lower than that in September. It suggests that under the weather conditions of





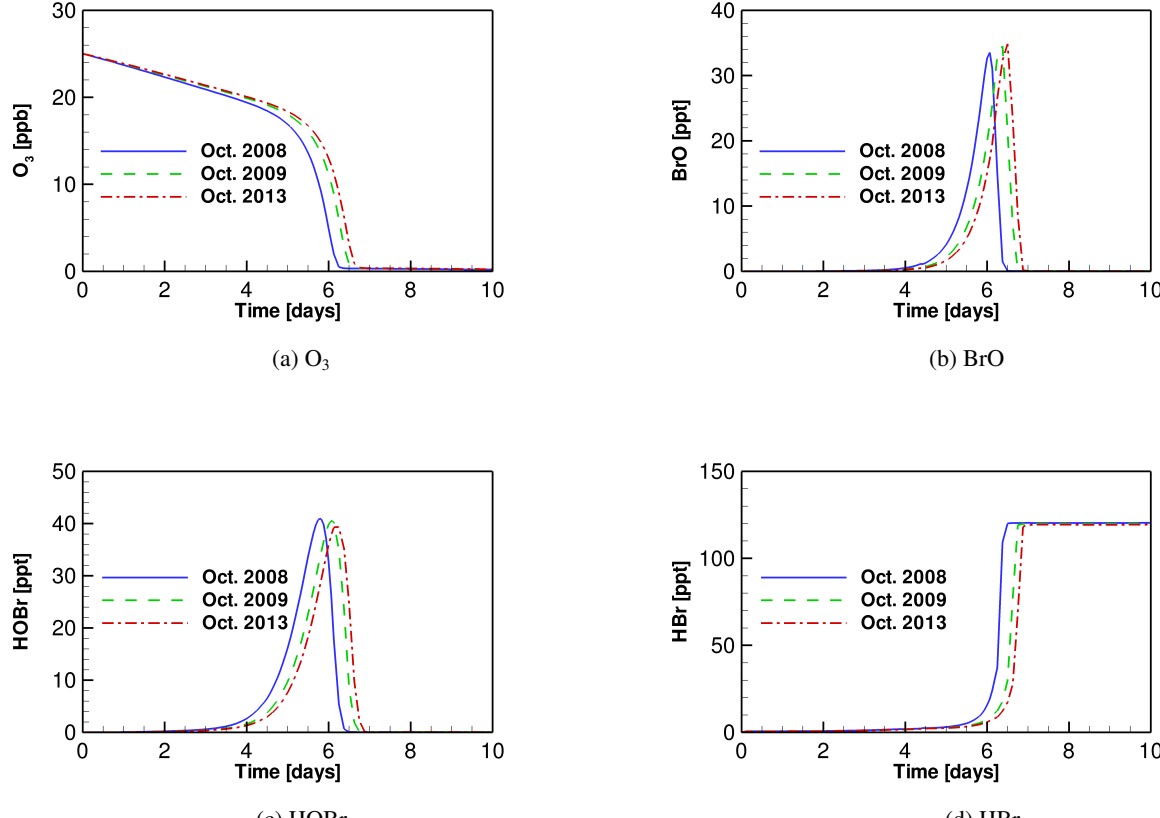

**Figure 5.** Variations of (a) ozone, (b) BrO, (c) HOBr and (d) HBr with time during ODEs under the conditions of October for three different years (i.e. 2008, 2009 and 2013). The TOC values implemented in the model for these three years are 145.5 DU, 169.2 DU and 179.9 DU, respectively.

November 2013, the occurrence of ODEs is more difficult to achieve, compared with that in September 2013. This result is also
consistent with the change in the occurrence frequency of ODEs shown above. However, as mentioned above, although TOC increases from September to November in 2013, it does not guarantee that the occurrence of ODEs is negatively dependent on the variation of TOC, because SZA also decreases, which may heavily enhance the radiation fluxes reaching the ground surface.

Therefore, we continued to test different TOC that corresponds to the October month for three different years (145.5 DU
for Oct. 2008, 169.2 DU for Oct. 2009, and 179.9 DU for Oct. 2013) in the models. In these simulation scenarios, the TOC value for the year 2008 is the lowest, while the TOC value for 2013 is the highest. In contrast, the values of SZA and other meteorological parameters are mostly similar in these scenarios. By doing that, we were able to figure out the impact on the occurrence of ODEs brought about only by the change in TOC using TUV and KINAL models.



The time variations of ozone and major bromine species are presented in Fig. 5. It can be seen in Fig. 5(a) that when TOC
in October is lower (i.e. the situation in 2008), the depletion of ozone is accelerated and the depletion rate becomes higher
than those for the other two years. It denotes that it takes less time for the ozone to be completely depleted in 2008 than in
the other two years due to the lower TOC in the October of 2008. Moreover, it can be found from Fig. 5(b)-(d) that the peaks
of BrO, HOBr and HBr in October 2008 occur earlier than those in October 2013. In addition, it can be seen from Fig. 5(d)
that the total amount of bromine in the atmosphere (i.e. in the form of HBr) at the end of ODEs for 2008 is slightly higher
than that for 2013. Thus, from the model results, the decrease of TOC favors the occurrence of the tropospheric ODEs and the
bromine release. This simulation result also partly supports our previous suggestion about the negative dependence of the ODE
occurrence frequency on the TOC change in the analysis of the observational data.

The mechanism we proposed is that when TOC decreases, a larger amount of solar radiation would reach the troposphere,
leading to an acceleration of photo-chemical reactions in the troposphere. As a result, the formation of major bromine species
such as BrO and HOBr as well as the bromine activation become faster, and the occurrence of ODEs is also accelerated.

However, it is still unclear through which photolysis reactions the change in TOC deeply affects the occurrence of tropospheric ODEs. Thus, we continued to analyze the photo-chemical reactions using the concentration sensitivity analysis, as
presented below.

### 3.4 Sensitivities of Ozone and Major Bromine Species to Photolysis Reactions

In the present study, the impact on ODEs caused by the change in TOC in the models is exerted through 23 photolysis reactions,
which are listed in Tab. 1. In order to figure out which photolysis reactions are the most important ones during the ODE process,
we performed a concentration sensitivity analysis on the change in ozone and major bromine species for the springtime of 2013,
so that the dependence of ozone and bromine species on these photolysis reactions can be revealed.

The relative concentration sensitivities of ozone and major bromine species (i.e. BrO, HOBr and HBr) to all the 23 photolysis
reactions on Day 6.5, which resides in the second time stage of ODEs when the strongest ozone depletion occurs (see Fig. 3),
are shown in Fig. 6. From Fig. 6(a), it can be seen that Reactions (SR7) and (SR11):

$$\text{BrO} + h\nu \xrightarrow{\text{O}_2} \text{Br} + \text{O}_3, \tag{SR7}$$

$$\text{HOBr} + h\nu \rightarrow \text{Br} + \text{OH}, \tag{SR11}$$

possess the largest positive sensitivities for the mixing ratio of ozone. It means that the rate increase of these two photolysis
reactions leads to an elevation of the ozone value during ODEs and thus a retardation of ODEs. These two reactions are thus
named "major ODE decelerating reactions" in the following context. The reason for the delay impact on ODEs brought by
Reaction (SR7) is that in this reaction, BrO is photolyzed, forming Br. As a result, the formation of HOBr by the oxidation of
BrO is decelerated due to the reduction of the available BrO. Thus, the heterogeneous bromine activation process, i.e. bromine
explosion mechanism that HOBr participates in, gets retarded, leading to a slow down of the bromine activation and the ozone







(a) $O_3$

(b) BrO

(c) HOBr

(d) HBr

**Figure 6.** Relative sensitivities of (a) ozone, (b) BrO, (c) HOBr and (d) HBr to photolysis reactions on Day 6.5, which resides in the time period when the strongest ozone depletion occurs.

depletion. Apart from that, additional ozone is also formed through Reaction (SR7). With respect to Reaction (SR11), HOBr is photo-decomposed through this reaction. Thus, the heterogeneous bromine activation also gets suppressed by the strengthening of this reaction, resulting in a delay of the ozone depletion.

In contrast, Reactions (SR1), (SR57), and (SR58),

$$O_3 + h\nu \rightarrow O(^1D) + O_2, \tag{SR1}$$





$$H_2O_2 + h\nu \rightarrow 2OH, \tag{SR57}$$

$$HCHO + h\nu \xrightarrow{2O_2} 2HO_2 + CO, \tag{SR58}$$

are largely negatively correlated with the change in ozone (see Fig. 6a). It denotes that when these three reactions speedup, the level of ozone drops, which represents an acceleration of the tropospheric ODE. These three reactions are thus named "major ODE accelerating reactions". For Reaction (SR1), it is not surprising as this reaction is the direct photolysis of the tropospheric ozone. Moreover, Reaction (SR1) also serves as a major formation pathway of OH radicals. OH radicals are important for the occurrence of ODEs and the bromine explosion mechanism, as they are involved in Reaction (SR18):

$$Br_2 + OH \rightarrow Br + HOBr. \tag{SR18}$$

In Reaction (SR18), not only the Br atoms are generated by the conversion from the released $Br_2$, but also the formation of HOBr is strengthened, leading to an acceleration of the bromine activation. As a result, an enhancement of Reaction (SR1) would lead to a speedup of the ozone depletion. Regarding Reaction (SR57), the occurrence of this reaction also strengthens the occurrence of ODEs, because this reaction also forms OH radicals, which are critical for the bromine explosion mechanism 330 as mentioned above. With respect to Reaction (SR58), it possesses the most negative ozone sensitivity (see Fig. 6a), which means that this reaction heavily controls the depletion of ozone. It is because this reaction reinforces the formation of $HO_2$. As $HO_2$ is the key oxidant for the formation of HOBr through reaction:

$$BrO + HO_2 \rightarrow HOBr + O_2, \tag{SR10}$$

the enhancement of Reaction (SR58) thus favors the release of bromine and the depletion of ozone.

From Fig. 6(b)-(d), we can see that the sensitivities of major bromine species (i.e. BrO, HOBr and HBr) to photolysis reactions mostly have an opposite sign, compared with those corresponding to ozone (shown in Fig. 6a). It is because the bromine species in the atmosphere are mostly responsible for the ozone depletion in the troposphere during ODEs. Within these photolysis reactions, Reactions (SR7) and (SR11) have the largest negative sensitivities as they strongly decelerate the bromine explosion mechanism. In contrast, Reactions (SR1), (SR57) and (SR58) exert a positive impact on the change in bromine 340 species. It is because the speedup of these three reactions can reinforce the bromine explosion mechanism as mentioned above, thus leading to a positive dependence of these bromine containing compounds on these three reactions.

    The results of the sensitivity analysis help to explain the impact on the occurrence of ODEs exerted by the change in TOC in previous discussions. First, in the comparison of the results corresponding to different months of the springtime of 2013 (see Sect. 3.3), it was found that the tropospheric ODE is more difficult to achieve in November than in September of the same 345 year. From the sensitivity analysis, we attributed the reason for the retardant of ODEs in November to be the smaller SZA in November than that in September. Due to the shift in SZA, solar radiation with all the wavelengths that reaches the ground





surface is strengthened in November. Consequently, although the TOC value in November is higher than that in September, both the major ODE accelerating reactions (i.e. Reactions (SR1), (SR57) and (SR58)) and the major ODE decelerating reactions (i.e. Reactions (SR7) and (SR11)) are promoted in November. From Fig. 6(a), it can be seen that the ozone level during ODEs

is more sensitive to the major ODE decelerating reactions than the major ODE accelerating reactions. As a result, the outcome of the SZA decline in November is that the occurrence of ODEs is retarded. In this situation, the change in TOC only plays a minor role in affecting the occurrence of ODEs.

In contrast, in the comparison of the ODE occurrence belonging to October of different years (see Sect. 3.3), because the values of SZA are mostly similar in these scenarios, the change in ODEs is mainly determined by the difference in TOC

between these simulations. In a lower TOC environment, the intensity of the solar radiation reaching the atmospheric boundary layer, especially the ultraviolet radiation in a wavelength range of 200-320 nm (i.e. UV-B and UV-C), is significantly enhanced, as ozone has strong absorption bands in 200-320 nm (i.e. Hartley bands). Moreover, solar radiation in 320-350 nm also gets moderately elevated under a low TOC condition, because of the absorption bands of ozone in 320-350 nm with a vibrational structure (i.e. Huggins bands). On the contrary, solar radiation in other wavelength ranges would not be significantly affected by

the decrease of TOC. In this situation, the major ODE accelerating reactions, i.e. photolysis of ozone, $H_2O_2$ and HCHO in the boundary layer, are remarkably promoted. The reasons are as follows: (1) The photolysis of ozone in the troposphere that forms $O(^1D)$ depends heavily on the strength of the solar radiation in 295-360 nm (Akimoto, 2016). Thus, the decrease in TOC would significantly accelerate the photolysis of ozone in the troposphere, i.e. Reaction (SR1). (2) The absorption cross section of $H_2O_2$ decreases monotonically from the wavelength of 190 nm to 350 nm (Vaghjiani and Ravishankara, 1989), which overlaps the

wavelength range that the TOC change strongly affects. Thus, the photo-decomposition of $H_2O_2$ (i.e. Reaction (SR57)) also gets strongly promoted when TOC decreases. (3) The absorption spectrum of HCHO spreads from 260 nm to 360 nm, with many vibrational structures (Rogers, 1990). As a result, the decline of TOC substantially enhances the photolysis of HCHO (i.e. Reaction (SR58)).

On the contrary, for the major ODE decelerating reactions, i.e. photolysis of BrO and HOBr, their rates are only slightly

influenced by the decrease of TOC. It is because that BrO has an absorption spectrum in the range of 290-380 nm, peaking at approximately 330 nm (Wilmouth et al., 1999). The strengthening of the solar radiation especially in 200-320 nm due to the decline of TOC thus only exerts a small impact on the photolysis of BrO (i.e. Reaction (SR7)). Regarding HOBr, its photolysis rate relies more on the strength of the UV-A radiation (i.e. in 320-400 nm) reaching the boundary layer, as it has a broad absorption spectrum between 200 and 400 nm (Burkholder et al., 2015). Therefore, the influence caused by the TOC change

on the photolysis of HOBr is also weak.

Hence, it can be concluded that when TOC decreases, the rates of major ODE accelerating reactions (i.e. Reactions (SR1), (SR57) and (SR58)) significantly increase, while the rates of major ODE decelerating reactions (i.e. Reactions (SR7) and (SR11)) are hardly changed. Consequently, the ODEs are accelerated under a low TOC condition, and vice versa, resulting in a negative association between the change of TOC and the occurrence frequency of ODEs as presented above.





## 4   Conclusions and Future Work

In this study, we investigated the connection between the change in the total ozone column (TOC) and the occurrence of the tropospheric ozone depletion events (ODEs) in the Antarctic. Photolysis reactions dominating this connection were also identified using a concentration sensitivity analysis. Major conclusions achieved in the present study are as follows.

Based on the analysis of the observational data belonging to the years 2007-2013, we suggested that the decrease in TOC possibly favors the occurrence of the tropospheric ODEs. Then, the model results with the input data representing the conditions of October belonging to different years indicate that the occurrence of ODEs would be accelerated when TOC decreases. Moreover, key photolysis reactions that dominate the production and the consumption of ozone during ODEs, i.e. major ODE accelerating reactions and major ODE decelerating reactions, were also figured out. It was found that when TOC changes, the rates of major ODE accelerating reactions are substantially altered, while the rates of major ODE decelerating reactions mostly remain unchanged, leading to the negative association between the TOC change and the occurrence frequency of ODEs.

Improvements can be made to the present study. For instance, many other factors that are able to influence the occurrence of ODEs such as the type of the sea ice and the existence of frost flowers should be considered in the future work. Unfortunately, currently we are still lack of these observational data. Aside from that, a study for Arctic conditions should also be conducted, so that the conclusions obtained in the present study can be compared and verified, which is attributed to a future publication.

*Code and data availability.* The source code of the model and the data of computational results shown in this article can be acquired upon request from the authors.

*Acknowledgements.* The authors wish to thank the financial support by the National Natural Science Foundation of China (Grant No. 41705103). The numerical calculations in this paper have been done on the high performance computing system in the High Performance Computing Center, Nanjing University of Information Science & Technology.

*Author contributions.* Le Cao conceived the idea of the article and extended the KINAL model. Linjie Fan processed the observational data, performed the computations, and wrote the paper with Le Cao together. Simeng Li revised the chemical mechanism and Shuangyan Yang gave valuable suggestions on the improvement of the manuscript. All the authors listed have read and approved the final manuscript.

*Competing interests.* The authors declare no conflict of interest.





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
