# Peer review of "Influence of Total Ozone Column (TOC) on the Occurrence of Tropospheric Ozone Depletion Events (ODEs) in the Antarctic"

_Atmospheric Chemistry and Physics, 2021_

## Referee Comment (RC2)

**Influence of the Change in Total Ozone Column (TOC) on the Occurrence of Tropospheric Ozone Depletion Events (ODEs) in the Antarctic by Cao et al.**

The article deals with the influence of TOC on ODEs. We had earlier studies on ODEs and the chemistry associated with them, such as bromine driven chemistry. The authors try to find the link between ozone amount and ODEs here. This is an interesting point of discussion. The MS is well-written and the model results are adequate to explain the associated chemistry. However, there are still some unclear points, which are need to be addressed. The MS can be published after this successful revision.

**Major:**

Until the description of modelling section there is this confusion that

1.  What is the mechanism that drives ODEs?
2.  What are the chemistry associated with this?
3.  Are you talking about ODEs or TOC decrease?

Therefore, a careful rephrasing is needed in some places to clarify these doubts.

**Minor:**

L 1-2: I do not understand this. TOC influences ODS or the other way around?

L 2: Usually "data" means measurements or observations (not model, reanalyses, etc.)

L 7: seems? So how it happens? Just changes in TOCs?

L 9-11: confusing .please state clearly

L24: an extraordinary

L 26: The measurements of TOC from these stations are described in detail by Kuttippurath et al., 2010. It's good to mention this here, so that the readers will get an idea about these station measurements.

Kuttippurath, J., Goutail, F., Pommereau, J.-P., Lefèvre, F., Roscoe, H. K., Pazmiño, A., Feng, W., Chipperfield, M. P., and Godin-Beekmann, S.: Estimation of Antarctic ozone loss from ground-based total column measurements, Atmos. Chem. Phys., 10, 6569–6581, https://doi.org/10.5194/acp-10-6569-2010, 2010.

L 33: it was first reported that

L 35; in the coastal

L 48: such as Temperature

L53: there is no observational evidence

L59: at the Arctic coastal stations

L 85: "figure out", use another word.

L 119: It depends, not for all stations. Please rephrase

L 121: What is out of scope, as long as Faraday station is in the study region?

L 125: not really, they have longer periods of data. See Kumar et al., 2021, who have used same the data from those stations

Kumar P., J. Kuttippurath, P. von Gathen et al.: The increasing surface and tropospheric ozone in Antarctica and their possible drivers, Environmental Science and Technology, https://doi.org/10.1021/acs.est.0c08491, 2021

L 136: What kind of extreme events?

L 154: reference for US standard atmosphere. How old is this reference table?

L207: "Later, we present .."

L277-278: "By doing that, we were able to figure out the impact on the occurrence of ODEs brought about only by the change in TOC using TUV and KINAL models."

TOC changes control the ODEs?

L285-286: "Thus, from the model results, the decrease of TOC favors the occurrence of the tropospheric ODEs and the bromine release."

TOC releases bromine? You need to explain this in terms of chemistry not with the TOC changes. TOC change can happen with dynamics too

Figure 2: There is an ant-correlation between TOC and ODEs, except for 2010, any reason?

In addition, you state that the ODEs accelerate when TOCs are smaller, not the other way around. However, the figure does not illustrate that (conclusions line 384-385). Please explain.

---

## Author Comment (AC1)

**Response to Referee #1**

The authors sincerely thank Reviewer #1 for the comments, which greatly contribute to an improvement of this paper.

In the following, we address the questions raised by the reviewer:

**Major comments:**

**Q1.1:** In the title and at numerous occasions in the manuscript the authors claim to study "the influence of the change in TOC" on ozone depletion events. However, the manuscript only shows the influence of the TOC on the depletion events. This is most obvious in Fig. 1, with a caption referring to "… temporal change in TOCs", while it shows time series of observed TOCs at different locations. Of course, the TOC varies over time. However, the authors should carefully check when they actually studied the influence of a changing TOC or when they studied different TOCs.

**A1.1:** Thanks a lot for pointing this issue out. Yes, we had some confusions about the difference between a changing TOC and different TOCs in the original manuscript. In the present study, we actually investigate the impact exerted by a changing TOC on ODEs. We have carefully revised our manuscript again, and made many corrections. Please see the contents marked in red (e.g., **lines 5-6, lines 86-88, line 97**) in the revised manuscript. The title of the paper is also modified according to the reviewer's suggestion. Thanks again for this valuable suggestion.

**Q1.2:** I find ch. 3.1 dealing with the observations not convincing. In the current form a relationship between the presented TOC and the ozone depletion near the surface is not apparent. However, why would you expect to find a strong relationship between a short-term phenomenon that lasts a couple of days at maximum and the monthly average of the TOC? Table 3 indicates that in the studied period at Halley the periods classified as depleted in ozone are always less than 10 % of the total time. In my opinion the monthly averages can easily hide any correlation. Moreover, the spatial representativeness of the two observations (surface ozone and TOC) are not discussed. Depending on the meteorological conditions the observed surface ozone may correspond to a very confined region, while it can also be influenced by the effective transport of air masses. In fact, it is well known that depletion events at coastal stations are in almost all cases related to the transport of air masses from sea ice-covered areas. This is not discussed at all in the manuscript. On the other hand, it remains unclear what a monthly average of TOC signifies in terms of spatial representativeness. I am not so familiar with the spatial and temporal variability of stratospheric ozone, but what does a monthly average of TOC represent? The use of monthly averages is even more surprising since it appears that higher resolution data are available. Why didn't the authors use daily values of the TOC and the surface ozone to check for any correlations?

**A1.2:** Thanks. In this comment, the reviewer doubted about the use of the monthly averaged TOC in the original manuscript. At first, we would like to state the reasons we used a monthly averaged TOC in the original manuscript. Yes, it is well known that the ODEs detected at a single monitoring station are mostly caused by a transport of ozone-lacking air from halogen-rich areas such as the sea covered by fresh ice. However, to our knowledge, the exact source region of the air mass for a single monitoring station is still unclear at present. Therefore, if we wish to investigate the relationship between the surface ozone observed at a monitoring station and the TOC detected at its source region, a lag time should be taken into account. However, this lag time is also unclear (2 days? 3 days? or longer), which heavily depends on the weather conditions of the studied period. That is why we used a monthly averaged TOC in the original manuscript and turned to investigate the correlation between this monthly averaged TOC and the occurrence of ODEs within each month. By doing that, we can possibly avoid considering the lag time between the variations of the surface ozone and the TOC.

However, after carefully reading the comment of the reviewer, we do feel that the treatment of the observational data especially using the monthly averaged TOC in the original manuscript is inappropriate. Thus, we collected more observational data including the daily TOCs detected at different monitoring stations in the Antarctic, and tried to find out the association between the surface ozone observed at Halley and the daily TOCs belonging to different stations. By doing that, we can also possibly find a source region of the ODEs detected at the Halley station. Therefore, in the revised manuscript, not only the daily TOC data from the Halley station, but also the daily TOC data from Faraday-Vernadsky (FAD) station (65.25 °S, 64.27 °W) were used for a further analysis. Moreover, in order to guarantee the representativeness of the TOC data from these two stations, we also compared the TOC recorded at Halley with that obtained at a station nearby, Belgrano II (77.88 °S, 34.63 °W). We found that the correlation coefficients between the TOCs obtained at the Halley station and the Belgrano II station mostly possess a value above 0.9 (see Tab. S1 in the revised supplementary material), indicating that the TOC obtained at the Halley station can represent the typical TOC variation surrounding Halley. In addition, we validated the observed TOC obtained at the FAD station using the observations from Marambio station (64.24 °S, 56.62 °W), which is located on the northeast side of the Antarctic Peninsula, and we found the correlation between these two data high, which ensures the validity of the TOC observed at the FAD station.

The daily TOCs detected at Halley and Faraday-Vernadsky (FAD) as well as the surface ozone observed at Halley during the springtime of years 2007-2013 are shown in Fig. A1 of this rebuttal. The ODEs identified in this study are also marked in this figure.

[Figure]

Fig. A1 (Continued…)

[Figure]

Fig. A1 Time series of TOCs belonging to the Halley station and the Faraday-Vernadsky (FAD) station as well as the surface ozone detected at Halley during the springtime of years 2007-2013 (the observational data of the surface ozone for October and November in the year 2012 are missing). The green-shaded areas in the figure indicate the periods identified as the occurrence of ODEs at Halley in the present study, and the red-shaded areas represent the significant decline in TOC at FAD, which might be associated with the occurrence of ODEs at Halley.

From Fig. A1, we found that the daily TOCs observed at these two stations (i.e., Halley and FAD) are different. We also calculated the correlation coefficients between TOCs observed at the Halley station and the FAD station, and the correlation coefficients mostly reside in a value range of 0.3-0.8. The difference between the observed TOCs at these two stations may be caused by atmospheric dynamics as pointed out by Reviewer #2. Moreover, by comparing the surface ozone of Halley with the TOC detected at Halley (see Fig. A1), we did not find any obvious correlation between them, except that the ODEs occur more frequently in a relatively low TOC condition.

However, from the comparison between the surface ozone of Halley and the TOC detected at the Faraday-Vernadsky (FAD) station, we found that the ODEs observed at Halley usually followed a decline of TOC detected at the FAD station (see the marks in Fig. A1). It suggests that the decrease of TOC over the area of FAD possibly favors the occurrence of ODEs at the Halley station. As the FAD station is located to the northwest of the Halley station and near the Weddell Sea (see Fig. A2 in this rebuttal for locations of the FAD station and the Weddell Sea), the TOC detected at this station is more capable of reflecting conditions of the Weddell Sea. Thus, we suggest the possible mechanism as that the decline in TOC over the area of the Weddell Sea favors the tropospheric ozone depletion in this region. Then the ozone-lacking air was transported from the sea to the Halley station, leading to the detection of ODEs at this site. Thus, there exists a lag time between the TOC decline observed at the FAD station and the detection of ODE at the Halley station, and the length of the lag time depends on the weather conditions during that period. In previous studies, the source of ODEs observed in Halley has also been discussed by by Jones et al. (2006), who found that air masses causing rapid ODEs of Halley originated in the southern Weddell Sea. Our findings are consistent with the conclusions of Jones et al. (2006).

[Figure]

Fig. A2 Locations of the Faraday-Vernadsky (FAD) station and the Weddell Sea in the Antarctic.

Due to the replacement of the monthly averaged TOC by the daily TOC observed at different stations in this study, different simulations and sensitivity tests were performed in the revised manuscript, which will be presented and discussed in details in a later context (answer **A1.3** in this rebuttal). We also modified the contents about the description of the observational data and the discovery of the relationship between TOCs and the surface ozone of Halley in the revised manuscript. Please see **Section 2.1.1** and **Section 3.1** for the refined results and related discussions. Thanks again for giving such a valuable suggestion, so that our paper improves a lot.

**References:**

Jones, A. E., Anderson, P. S.,Wolff, E.W., Turner, J., Rankin, A. M., and Colwell, S. R.: A role for newly forming sea ice in springtime polar tropospheric ozone loss? Observational evidence from Halley station, Antarctica, Journal of Geophysical Research: Atmospheres, 111, 2006.

**Q1.3:** I find the modeling part of the manuscript interesting and useful. To my knowledge, the influence of the TOC on the depletion of surface ozone via its impact on the different photolysis rates has not been studied in such detail before. This gives useful new information on the processes governing the depletion of ozone. This also concerns the sensitivities as shown in ch. 3.4. Such information may also be helpful to understand why ozone depletions only appear during springtime and what processes are involved in the termination of depletion events. However, I am less convinced by the choice of the boundary conditions that are used for the simulations. It appears that the monthly averages of the TOC as presented in ch. 3.1 were used as well as a specific solar zenith angle SZA for each month. What is this SZA? The maximum, minimum, or average SZA for a given month or the SZA for the middle of the month? Was the diurnal cycle of the SZA considered in the simulations? Like in the case of the monthly averaged TOC for the correlations, I am not convinced that applying monthly values is useful and may even create artificial boundary conditions. Why didn't the authors perform simulations for specific days (for example, for each analyzed month the days with the maximum and minimum TOC or selected days with similar TOC, but varying SZA)? In my opinion such simulations would be much more convincing since they would more closely correspond to conditions encountered at Halley. Moreover, they would be easier to characterize and also easier to reproduce.

**A1.3:** First, we would like to thank reviewer for the praise on the modeling work of this study. Second, we agree with the reviewer that using a monthly averaged TOC in the TUV model is inappropriate, as the physical meaning of the monthly averaged TOC is not clear enough. Thus, as mentioned above, in the revised manuscript, we replaced the input of the model by a varying daily TOC for the present investigation, and then compared the model results for different months of the springtime of 2008. The adopted periods corresponding to the three months for the present investigation are Sept. 1-Sept. 10, Sept. 29-Oct. 8 and Nov. 1-Nov. 10, respectively. The temporal profiles of ozone

during ODEs under the conditions of different months of 2008 and the time series of TOC during these periods are shown in Fig. A3 of this rebuttal.

[Figure]

Fig. A3 Temporal profiles of TOC in different months of the springtime of 2008 and the simulated surface ozone during ODEs.

From Fig. A3, it is seen that compared with the situation in September, the decline of ozone in November is delayed. Moreover, the depletion rate in November was also found to be lower than that in September. It suggests that under the weather conditions of November 2008, the occurrence of ODEs is more difficult to achieve, compared with that in September 2008. However, although TOC differs between these months, it does not guarantee that the difference in the ODE occurrence between these simulations is caused by the use of different TOCs, because SZA also varies, which may heavily influence the radiation fluxes reaching the ground surface. We thus designed two new sensitivity tests in the present study, to discover the role of the solar zenith angle (SZA) and TOC in affecting the ODEs.

One sensitivity test added in the revised manuscript is that the input TOC variation in September and November simulations is replaced with that belonging to October. By performing this test, we were able to discover the influence on the occurrence of ODEs solely by SZA. Figure A4 in this rebuttal shows the temporal evolution of the surface ozone in this sensitivity test. It can be seen that when applying a same TOC variation in these simulations, the ozone depletion in September occurs remarkably earlier than that in November. It denotes that the decline of SZA leads to a retardation of ODEs. Thus, the ozone depletion in the troposphere is more difficult to achieve when SZA becomes smaller.

[Figure]

Fig. A4 Temporal evolution of ozone during ODEs in different months of the springtime of 2008, using a same TOC temporal profile.

The other sensitivity test is that we implemented different temporal profiles of TOC in the October simulation. In the original simulation for October, TOC observed at the FAD station drops sharply from 250DU in Sept. 29 to 131DU in Oct. 4. In this sensitivity test, we assumed that the TOC keeps as a constant 250DU or increases sharply from 250DU to 350DU instead of dropping, and we named these two simulation scenarios as the "constant" scenario and the "increase" scenario, respectively. Thus, in these simulation scenarios (original, "constant" and "increase"), the TOC variations are different, while the values of SZA and other input meteorological parameters are similar. By doing that, we were able to find out the impact on the occurrence of ODEs brought about only by TOC. The temporal profiles of ozone in this sensitivity test are presented in Fig. A5 of this rebuttal.

[Figure]

Fig. A5 Temporal evolution of ozone during ODEs under the conditions of October 2008, implementing three different TOC profiles (i.e., original, "constant" and "increase").

It can be seen in Fig. A5 that when TOC in October is lower (i.e., the situation in the original simulation), the depletion of ozone is accelerated and the depletion rate becomes higher than those in the other two scenarios. It denotes that it takes less time for the ozone to be completely depleted under a lower TOC. Thus, from the model results, the decrease of TOC favors the occurrence of the tropospheric ODEs.

After conducting these new simulations, we were able to clarify the role of SZA and TOC in affecting ODEs. The results of these new simulations and the related discussions are also added into the revised manuscript. Please see **lines 320-347**.

**Q1.4:** A further issue that merits some discussion is the comparison between the observed and the simulated surface ozone concentrations. First, it would be good to have a figure (maybe in the supplement?) showing the surface ozone concentrations that were used and indicating also the periods that were identified as periods with depleted ozone according to Table 3. Second, with the low frequencies as shown in Table 3 (the highest value corresponds to less than 70h during a full month) it is impossible that any of the events corresponds to the simulations as shown for example in Fig. 3 with three full days of zero ozone. This point should be clarified.
**A1.4:** First, according to the reviewer's comment, we added **Fig. 1** (i.e., Fig. A1 in this rebuttal) showing the temporal variation of the surface ozone observed at Halley into the revised manuscript. Periods identified as the occurrence of ODEs are also marked

in the figure.

The reviewer also questioned the method identifying ODEs. In the present study, the following criterion is used to indicate the occurrence of ODEs:

$$[O_3]_i - \overline{[O_3]} < -\alpha \cdot \sigma \qquad (1)$$

In Eq. (1), $[O_3]_i$ is the instantaneous ozone at the $i$-th time point, and $\overline{[O_3]}$ is the mean ozone value over a month. $\sigma$ in Eq. (1) is the standard deviation, and $\alpha$ is a constant, which is set to 1.5 in this study. Based on Eq. (1), we defined the occurrence of ODEs as the period when the ozone concentration drops remarkably instead of the period when the ozone level is lower than a criterion. Due to this definition, the time representing the occurrence of ODEs in this study is relatively shorter than that indicated in previous studies. We have added more explanations about the identification of the occurrence of ODEs in the revised manuscript. Please see **lines 164-168**. Thanks.

Moreover, as we currently use the daily value of the surface ozone, we refined the selection criterion of ODEs used in this study. The constant $\alpha$ in Eq. (1) is set to 1.5 instead of 2.0 so that many partial ODEs can also be identified. The identified ODEs using this selection criterion are also shown in Fig. A1 of this rebuttal, and from these results we feel the method identifying ODEs used in the present study acceptable.

**Q1.5:** The authors claim that they obtained the surface ozone measurements at Halley from the WDCGG. However, the WDCGG web page states: "Reactive gases measurement data (except for CO) have been agreed to be transferred under the responsibility of the newly established GAW World Data Centre for Reactive Gases (WDCRG) hosted by the Norwegian Institute for Air Research (NILU)." The source of the data should be verified.

**A1.5:** Thanks. We checked the source of the observational data again. We actually obtained the surface ozone data from the "legacy" section in the WDCGG web page https://gaw.kishou.go.jp/publications. However, during the revision process of this manuscript, we found that WDCGG does not provide these archived data any more. As the reviewer pointed out, these data have been transferred to WDCRG. We then compared the surface ozone data of Halley provided by WDCRG with the original data we obtained from WDCGG, and found these two data consistent.

We refined the statement about the source of the surface ozone data of Halley in the revised manuscript. Please see **lines 144-148** in the revised manuscript. We also added these data into the **Section "Code and data availability"**. Thanks a lot for the reviewer's comment.

**Q1.6:** The information about the parameters used in the different model components is not adequate since their description is too limited to support the reproducibility. For example, the TUV model (ch. 2.2.1) requires a range of further input parameters concerning the albedo, clouds, and aerosols. In the manuscript it is not sufficiently specified, which parameters were used. Moreover, according to equation (4) the

KINAL model (ch.2.2.2) does not include deposition, which can be an important removal process for a number of the simulated species. This would be a serious weakness of the model.

**A1.6:** Thanks a lot for pointing it out. We added more details of the TUV model input in **Tab. S2** in the revised supplements so that the readers can reproduce the model results. Please also see the explanation added into the revised manuscript, which is marked in red in **lines 185-187** of the revised manuscript.

Regarding the deposition process, in previous box model studies (e.g., Michalowski et al., 2000), it is usually assumed that the loss of chemical species caused by dry deposition is equivalent to the flux caused by the entrainment from the free atmosphere into the boundary layer. By doing that, in the absence of chemistry, the concentrations of chemical species are able to maintain in the balance of the dry deposition and the entrainment. As we did not include the flux caused by the entrainment from the free troposphere in the present model, we thus did not include the deposition of many chemical species either, so that the balance can be kept. However, it should be mentioned that we do consider the dry deposition of HOBr on the ground surface. It is because the rates of the important heterogeneous reactions on the ice-/snow-covered surfaces (e.g., $HOBr + H^+ + Br^- \rightarrow Br_2 + H_2O$) are estimated based on the dry deposition rate of HOBr. The rates of the heterogeneous reactions with the involvement of HOBr are calculated as follows:

$$\frac{d[HOBr]}{dt} = -v_d/L \cdot [HOBr] = -(r_a + r_b + r_c)^{-1}/L \cdot [HOBr] \qquad (2)$$

In Eq. (2), [HOBr] is the HOBr concentration in the boundary layer. $v_d$ is the dry deposition velocity of HOBr at the ice/snow surface. $L$ is the boundary layer height. The estimation of the dry deposition velocity, $v_d$, depends on the values of three resistances, $r_a$, $r_b$ and $r_c$. Details of the parameterization of the HOBr dry deposition at the surface can also be found in our previous publications (e.g., Cao et al., 2014; Cao et al., 2016).

We added more details about the treatment of the dry deposition process in the present model, please see **lines 214-219** and **lines 222-226** of the revised manuscript.

**References:**

Cao, L.; Sihler, H.; Platt, U.; Gutheil, E. Numerical analysis of the chemical kinetic mechanisms of ozone depletion and halogen release in the polar troposphere. Atmos. Chem. Phys., 14, 3771–3787, 2014.

Cao, L.; Platt, U.; Gutheil, E. Role of the boundary layer in the occurrence and termination of the tropospheric ozone depletion events in polar spring. Atmos. Environ., 132, 98–110, 2016.

Michalowski, B. A., Francisco, J. S., Li, S.-M., Barrie, L. A., Bottenheim, J. W., and

Shepson, P. B., A computer model study of multiphase chemistry in the Arctic boundary layer during polar sunrise, J. Geophys. Res., 105( D12), 15131– 15145, doi:10.1029/2000JD900004, 2000.

**Minor comments:**

**Q1.7:** I'm not convinced that the direct transport of tropospheric ozone into the boundary layer has ever been demonstrated (l. 77ff). In any case, Kuang et al, 2017, reported only that an entrainment from the stratosphere occurred into the free troposphere at an altitude above 3000m.

**A1.7:** Yes, the citation and the related statement here are inappropriate. The impact on ODEs in polar regions caused by the ozone entrainment from the stratosphere has not been reported yet. Thus, we removed the related sentences here, see **lines 89-91** of the revised manuscript. However, in mid-latitude areas, researchers do find an ozone entrainment from the stratosphere into the boundary layer, thus affecting the surface ozone (see Langford et al., 2012 and Lin et al., 2012 for example).

**References:**

Langford A O, Brioude J, Cooper O R, et al., Stratospheric influence on surface ozone in the Los Angeles area during late spring and early summer of 2010[J]. J. Geophys. Res., 117: D00V06. DOI:10.1029/2011JD016766, 2012.

Lin M Y, Fiore A M, Cooper O R, et al., Springtime high surface ozone events over the western United States: Quantifying the role of stratospheric intrusions[J]. J. Geophys. Res., 117: D00V22. DOI:10.1029/2012JD018151, 2012.

**Q1.8:** In figure 4 it appears that the ozone concentrations in the simulations for October and November do not drop to zero. Is this realistic?

**A1.8:** Although the original figure has been changed in the revised manuscript so that this doubt does not exist, we still like to discuss this issue with the reviewer. In our opinion, ozone in simulations do not drop to zero due to the lack of active bromine in the air during that time. When ozone drops to a low value during ODEs, the formation of BrO becomes weak, which also suppresses the formation of HOBr. As HOBr is the key species for the bromine explosion mechanism, only a few bromine in the substrates can then be activated and released into the air under this condition. In contrast, HBr is continuously formed due to the scavenging of bromine by aldehydes, leading to the loss of active bromine in the atmosphere. As a result, the total amount of the active bromine in the air becomes so small that the ozone consumption by bromine is weak at that time. Meanwhile, nitrogen oxides emitted from the ground surface (snowpack/fresh-ice) can form ozone in the presence of solar radiation. As a result, the loss of ozone caused by bromine and the formation of ozone due to the photolysis of nitrogen oxides attain a balance so that the ozone concentration does not drop any further and keeps stable at a low level.

---

## Author Comment (AC2)

**Response to Referee #2**

The authors sincerely thank Reviewer #2 for the valuable comments and the very helpful considerations, which greatly contribute to an improvement of our paper.

In the following, we address the particular issues raised by Reviewer #2:

**Major:**

**Q2.1:** Until the description of modelling section there is this confusion that
1. What is the mechanism that drives ODEs?
2. What are the chemistry associated with this?
3. Are you talking about ODEs or TOC decrease?
Therefore, a careful rephrasing is needed in some places to clarify these doubts.
**A2.1:** Thanks a lot for the comment. To clarify the confusion of Reviewer #2, we made the following modifications to the original manuscript.
1. The mechanism driving ODEs is that in the presence of sunlight, the bromide ions stored in substrates such as ice-/snow-covered surfaces are activated by HOBr molecules in the atmosphere and then released into the atmosphere. The elevated bromine in the atmosphere then depletes the surface ozone in polar regions. We added more details in the Section "Introduction" to explain the mechanism driving ODEs. Please see **lines 44-45** and **lines 54-56** of the revised manuscript.
2. We actually included the description of the chemistry associated with ODEs in the original manuscript. Please see **lines 43-54**. In the revised manuscript, we added more explanations so that the readers can understand the chemistry more easily.
3. We distinguish them (ODEs and TOC decrease) more clearly in the revised manuscript. Please see **lines 62, 65 and 81**. Thanks a lot for the advice.

**Minor:**

**Q2.2:** L 1-2: I do not understand this. TOC influences ODES or the other way around?
**A2.2:** What we meant to express is that TOC influences ODEs. We have refined this sentence in the abstract. Thanks.

**Q2.3:** L 2: Usually "data" means measurements or observations (not model, reanalyses, etc.)
**A2.3:** Yes, the sentence here is not clear enough. We rephrased the related sentence. Please see **lines 2-4** in the revised manuscript.

**Q2.4:** L 7: seems? So how it happens? Just changes in TOCs?
**A2.4:** We used the word "seems" here because the conclusion is not definite according to the existing data. We have rephrased this sentence in the revised manuscript. Moreover, TOC is only one of the factors that can influence ODEs. To clarify the

confusion of the reviewer, we modified the expression here; please see **lines 7-10** in the revised manuscript.

**Q2.5:** L 9-11: confusing .please state clearly
**A2.5:** Modified. Please see **lines 12-15**. Thanks for pointing it out.

**Q2.6:** L24: an extraordinary
**A2.6:** Corrected.

**Q2.7:** L 26: The measurements of TOC from these stations are described in detail by Kuttippurath et al., 2010. It's good to mention this here, so that the readers will get an idea about these station measurements.
Kuttippurath, J., Goutail, F., Pommereau, J.-P., Lefèvre, F., Roscoe, H. K., Pazmiño, A., Feng, W., Chipperfield, M. P., and Godin-Beekmann, S.: Estimation of Antarctic ozone loss from ground-based total column measurements, Atmos. Chem. Phys., 10, 6569–6581, https://doi.org/10.5194/acp-10-6569-2010, 2010.
**A2.7:** We added the reference and related discussions in the revised manuscript according to the suggestion of the reviewer. Please see **lines 35-36**.

**Q2.8:** L 33: it was first reported that
L 35; in the coastal
L 48: such as Temperature
L53: there is no observational evidence
L59: at the Arctic coastal stations
L 85: "figure out", use another word.
**A2.8:** All done. Thanks

Q2.9: L 119: It depends, not for all stations. Please rephrase
**A2.9:** The related contents have been modified in the revised manuscript. Currently, we use the TOC observations provided by two surface monitoring stations (i.e., the Halley station and the Faraday-Vernadsky station). Data from other stations nearby (Belgrano II and Marambio) were also adopted to ensure the representativeness of the TOC data from these two stations. We found that the correlation coefficient between the daily TOCs observed at the Halley station and the FAD station resides in a value range of 0.3-0.8. The difference between the observed TOCs at these two stations might be caused by atmospheric dynamics.

Please see **lines 126-140** in the revised manuscript. Thanks

**Q2.10:** L 121: What is out of scope, as long as Faraday station is in the study region?
**A2.10:** The content here has been modified. Please see **lines 137-140**. Thanks.

**Q2.11:** L 125: not really, they have longer periods of data. See Kumar et al., 2021, who

have used same the data from those stations

Kumar P., J. Kuttippurath, P. von Gathen et al.: The increasing surface and tropospheric ozone in Antarctica and their possible drivers, Environmental Science and Technology, https://doi.org/10.1021/acs.est.0c08491, 2021

**A2.11:** We sincerely thank the reviewer for providing this information. According to this suggestion, we found more data of surface ozone from observation stations in the Antarctic, which can be used for a further investigation. Please see the modifications marked in red in **lines 144-148** of the revised manuscript. The reference recommended by the reviewer is also cited in the revised manuscript.

Q2.12: L 136: What kind of extreme events?

**A2.12:** We modified the expression here and added more explanations about the selection criterion used in this study. Please see **line 157** of the revised manuscript. Thanks.

Q2.13: L 154: reference for US standard atmosphere. How old is this reference table?

**A2.13:** We added the reference here. Please see **line 180**. Thanks.

Q2.14: L207: "Later, we present .."

**A2.12:** Corrected.

Q2.15: L277-278: "By doing that, we were able to figure out the impact on the occurrence of ODEs brought about only by the change in TOC using TUV and KINAL models."
TOC changes control the ODEs?

**A2.15:** TOC is only one of the factors affecting the occurrence of ODEs. The dominant mechanism that controls ODEs is actually the consumption by the massive bromine in the air due to the bromine explosion mechanism.

We modified the statement here in the revised manuscript. Please see **lines 334-336**.

Q2.16: L285-286: "Thus, from the model results, the decrease of TOC favors the occurrence of the tropospheric ODEs and the bromine release."
TOC releases bromine? You need to explain this in terms of chemistry not with the TOC changes. TOC change can happen with dynamics too

**A2.16:** The mechanism we derived in this study is that the decrease of TOC results in an increase of solar radiation especially the UV radiation reaching the ground surface. As a result, the rates of many photolysis reactions increase. As these photolysis reactions are important for the bromine release and the corresponding ozone depletion, the acceleration of these photolysis reactions thus lead to a speedup of the tropospheric ODEs. That is why we stated here that the decrease of TOC favors the occurrence of ODEs and the bromine release.

To clarify the confusion of the reviewer further, we added more explanations in the

revised manuscript. Please see **lines 344-351**. Thanks.

Q2.17: Figure 2: There is an ant-correlation between TOC and ODEs, except for 2010, any reason?
In addition, you state that the ODEs accelerate when TOCs are smaller, not the other way around. However, the figure does not illustrate that (conclusions line 384-385). Please explain.

**A2.17:** According to the suggestion of Reviewer #1, we currently investigate the correlation between TOC and the occurrence of ODEs based on the **daily** values of TOC. As a result, the content mentioned here is largely modified in the revised manuscript. Please see the statements in **lines 253-268** of the revised manuscript.

The daily TOCs detected at Halley and Faraday-Vernadsky (FAD) as well as the surface ozone observed at Halley during the springtime of years 2007-2013 are shown in Fig. A1 of this rebuttal. The ODEs identified in this study are also marked in this figure.

[Figure]

Fig. A1 (Continued…)

[Figure]

Fig. A1 Time series of TOCs belonging to the Halley station and the Faraday-Vernadsky (FAD) station as well as the surface ozone detected at Halley during the springtime of years 2007-2013 (the observational data of the surface ozone for October and November in the year 2012 are missing). The green-shaded areas in the figure indicate the periods identified as the occurrence of ODEs at Halley in the present study, and the red-shaded areas represent the significant decline in TOC at FAD, which might be associated with the occurrence of ODEs at Halley.

From Fig. A1, we found that the daily TOCs observed at these two stations (i.e., Halley and FAD) are different. We also calculated the correlation coefficients between TOCs observed at the Halley station and the FAD station, and the correlation coefficients mostly reside in a value range of 0.3-0.8. The difference between the observed TOCs at these two stations may be caused by atmospheric dynamics. Moreover, by comparing the surface ozone of Halley with the TOC detected at Halley (see Fig. A1), we did not find any obvious correlation between them, except that the ODEs occur more frequently in a relatively low TOC condition. However, from the comparison between the surface ozone of Halley and the TOC detected at the Faraday-Vernadsky (FAD) station, we found that the ODEs observed at Halley usually followed a decline of TOC detected at the FAD station (see the marks in Fig. A1). It suggests that the decrease of TOC over the area of FAD possibly favors the occurrence of ODEs at the Halley station. As the FAD station is located to the northwest of the Halley station and near the Weddell Sea (see Fig. A2 in this rebuttal for locations of the FAD station and the Weddell Sea), the TOC detected at this station is more capable of reflecting conditions of the Weddell Sea. Thus, we suggest the possible mechanism as that the decline in TOC over the area of the Weddell Sea favors the tropospheric ozone depletion in this region. Then the ozone-lacking air was transported from the sea to the Halley station, leading to the detection of ODEs at this site. Thus, there exists a lag time between the TOC decline observed at the FAD station and the detection of ODE at the Halley station, and the length of the lag time depends on the weather conditions during that period. In previous studies, the source of ODEs observed in Halley has also been discussed by by Jones et al. (2006), who found that air masses causing rapid ODEs of Halley originated in the southern Weddell Sea. Our findings are consistent with the conclusions of Jones et al. (2006).

[Figure]

Fig. A2 Locations of the Faraday-Vernadsky (FAD) station and the Weddell Sea in the Antarctic.

Due to the replacement of the monthly averaged TOC by the daily TOC observed at different stations in this study, different simulations and sensitivity tests were performed in the revised manuscript. We also modified the contents about the description of the observational data and the discovery of the relationship between TOCs and the surface ozone of Halley in the revised manuscript. Please see **Section 2.1.1** and **Section 3.1** for the refined results and related discussions.

**References:**

Jones, A. E., Anderson, P. S.,Wolff, E.W., Turner, J., Rankin, A. M., and Colwell, S. R.: A role for newly forming sea ice in springtime polar tropospheric ozone loss? Observational evidence from Halley station, Antarctica, Journal of Geophysical Research: Atmospheres, 111, 2006.

---

## Author Response (AR2)

**Response to Review #1**

The authors like to thank Reviewer #1 for revising our manuscript again. In the following, we address the issues raised by Reviewer #1.

**Q1.1:** Nevertheless, the entire manuscript would largely benefit from a careful proof-reading by a native English speaker since it still suffers from some oversimplified representations of complex processes, a number of misleading formulations, and in some places from confusing or imprecise expressions. Here are just three examples from the introduction:

L. 16 "… protecting the lives on the earth." Whose lives? It appears that a number of complex processes in the Earth system are summarized in just this one sentence.

L. 17f: Maybe better to write "At high concentrations it causes eye irritations and disorders of the lung function of human beings (Lippmann, 1991)."?

L20f: Not clear to me how vertical convection itself can be a source of tropospheric ozone since it just moves tropospheric ozone from one location to another?

**A1.1:** Thanks a lot. We have revised our manuscript again and made the corresponding changes according to the suggestion. Typesetting and language copy-editing will also be performed by the Copernicus Publications editorial office during the production of this paper.

**Q1.2:** Well before Bottenheim et al. 2009, it was for example shown by Kaleschke et al. 2006 and Jacobi et al. 2006 that bromine release and the depletion of tropospheric ozone can be related to new ice formation.

Kaleschke, L., et al., Frost flowers on sea ice as a source of sea salt and their influence on tropospheric halogen chemistry, Geophys.Res.Lett. 31, L16114, doi: 10.1029/2004GL020655, 2004.

Jacobi, H.-W., et al., Observation of a fast ozone loss in the marginal ice zone of the Arctic Ocean, J.Geophys.Res. 111, D15309, doi: 10.1029/2005JD006715, 2006.

**A1.2:** Thanks for the recommendation of references. We have added these two references and the related discussions into the revised manuscript. Please see **lines 71-74** in the revised manuscript.

**Q1.3:** Since the manuscript deals with tropospheric and stratospheric ozone, which is still easily confounded, the authors should carefully indicate throughout the manuscript if they talk about tropospheric or stratospheric ozone.

**A1.3:** Thanks for the suggestion. We indicated them more clearly in the revised manuscript.